# Procoagulant Effects of *Bothrops diporus* Venom: Kinetic Modeling and Role of Serine Protease Activity

**DOI:** 10.3390/ijms26199496

**Published:** 2025-09-28

**Authors:** Gisela L. Lopez, Sarah A. Nielsen, Vance G. Nielsen, Luciano S. Fusco

**Affiliations:** 1Consejo Nacional de Investigaciones Científicas y Técnicas (CONICET), Instituto de Química Básica y Aplicada del Nordeste Argentino (IQUIBA-NEA), Corrientes CP3400, Argentina; giselopezyaccuzzi@exa.unne.edu.ar (G.L.L.); luciano.fusco@comunidad.unne.edu.ar (L.S.F.); 2Department of Ecology and Evolutionary Biology, College of Science, University of Arizona, Tucson, AZ 85724, USA; sarahn2@arizona.edu; 3Department of Anesthesiology, College of Medicine, University of Arizona, Tucson, AZ 85750, USA; 4Facultad de Ciencias Exactas Naturales y Agrimensura, Universidad Nacional del Nordeste, Corrientes CP3400, Argentina

**Keywords:** *Bothrops* species, venom, serine protease, coagulation time, coagulation kinetics, spectrophotometry, thromboelastography, calcium dependent activity, antithrombin, ruthenium antivenom

## Abstract

*Bothrops* species are responsible for the majority of envenomations in Argentina. In particular, *Bothrops diporus* is among the main species responsible for the majority of envenomations in Argentina and causes significant injury and coagulopathy. Given the significance of this venom, the authors sought to define the toxin responsible for coagulopathy with specialized spectrophotometric and thromboelastographic methods. Utilizing clotting time, spectrophotometry, and thromboelastography, it was determined that *B. diporus* venom has potent, procoagulant activity in human plasma and buffer milieu. Calcium-dependent and -independent activities consistent with serine protease activity were identified. The activity included both thrombin-generating and thrombin-like enzymatic activity. The venom cleaved the serine protease-specific chromogenic substrate β-Ala-Gly-Arg-p-nitroanilide diacetate, and its activity was inhibited in plasma by antithrombin after addition of heparin. Further, venom exposed in isolation to RuCl_3_, a known inhibitor of serine protease-containing venoms, demonstrated decreased activity in human plasma. In conclusion, the present study contributes to a better understanding of *B. diporus* venom and may have implications for the rational design of inhibitors, antivenom formulations, or preclinical models to study venom-induced coagulopathies.

## 1. Introduction

Envenomation by *Bothrops* species vipers results in significant human morbidity and mortality in South America [1,2,3,4,5,6,7,8,9,10,11,12,13,14,15,16,17,18,19,20,21,22,23,24,25,26], and this genus accounts for the majority of venomous snake bites [1,2,3,4,5]. *Bothrops* envenomation is characteristically associated with severe, progressive tissue injury at and extending from the bite site [1,2,3,4,5,6,7], driven by cytotoxicity [27], and the inability to regenerate muscle if the venom is still present [28]. Further, *Bothrops* envenomation can inflict injury to organs remote from the bite site, injuring the heart [8], lungs [9], liver [9,10], and kidneys [9,11,12,13]. Neurological injuries that can occur include hemorrhagic/ischemic stroke [14,15,16,17,18,19,20] and reversible posterior leukoencephalopathy [21]. The likely mediator of these remote organ injuries is a procoagulant venom that can cause thrombosis within the vasculature, and in support of this, thrombotic microangiopathy has been observed following *Bothrops* envenomation [22,23,24]. Consumptive coagulopathy documented by several clinical hematological methods has been described [8,25,26] following envenomation. When considered as a whole, scenarios involving local and remote microvascular injury inflicted by *Bothrops* venom coupled with either tissue thrombosis or coagulopathy would be expected to result in ischemic or hemorrhagic outcomes as described [1,2,3,4,5,6,7,8,9,10,11,12,13,14,15,16,17,18,19,20,21,22,23,24,25,26]. Thus, further investigation of *Bothrops* venoms to define the molecular mechanisms that serve as the underpinning of toxicity continues to be of intense interest and focus.

Within the past few years, two groups have extensively focused on the effects of *Bothrops* venoms on human coagulation, using not just standard hematological methods but also with the use of thromboelastography [29,30,31,32]. Fry’s group in Australia has documented both the procoagulant and anticoagulant toxicities of numerous *Bothrops* species in human plasma, with a focus on several factors such as evolution [29,30]. Earlier work by Nielsen’s group also documented the primarily procoagulant effects of *Bothrops* venoms in human plasma [31,32]; however, the region from which the venom was collected was found to have marked effects on the degree of procoagulation by *Bothrops asper* [32]. Of particular interest, the thromboelastographic footprint of *Bothrops diporus* venom was documented, via thromboelastography in the presence of calcium, to be a weak–moderate procoagulant venom in human plasma, with no ability to polymerize pure fibrinogen [30]. The source of the venom was stated to be Venom Evolution Lab, UQ, Australia; the venom came from adult snakes, but the precise locality of origin was unknown, and it was unknown whether the venom came from an individual viper or was part of a pooled sample [30]. The thromboelastographic footprint of *B. diporus* [30] is consistent with serine protease-containing venoms that either act as thrombin or generate thrombin without the ability to directly polymerize fibrinogen as a thrombin-like enzyme (TLE) [32]. While these preliminary thromboelastographic findings were valuable [32], given the extensive range *B. diporus* has within Argentina and its marked hemotoxicity and myotoxicity [33], coupled with potential geographical variation in its procoagulant effect, as seen with *B. asper* [32], additional hematological investigation characterizing this venom is justified.

While isolation and characterization of phospholipase A_2_ within *B. diporus* venoms have been performed [34,35,36], detailed thromboelastographic and basic biochemical characterizations of serine proteases within *B. diporus* venom remain to be accomplished. While knowing the relative abundance of enzyme types in a venom proteome is critical to analyses of its potential effects in vitro and in vivo, the kinetics of catalysis of target molecules by specific enzymes in the bite site, circulation, or remote locations cannot be discerned from relative abundance. The relative abundances of metalloproteinases, serine proteases, and PLA_2_ in the venom of *Crotalus atrox*, *Crotalus adamanteous*, and *B. diporus* are displayed in Table 1 for comparison [37,38,39].

Specifically, if the Michaelis constant (Km) of a snake venom enzyme (or any other enzyme) is small and the maximum rate of reaction (Vmax) is large, then it will outcompete other venom enzymes present that catalyze the same substrate, resulting in a characteristic footprint in terms of assessment parameters. Thus, venoms kinetically predominated by fibrinogenolytic metalloproteinases [40,41] will cause progressive delays in the onset of coagulation, decreased velocity of growth, and decrease in maximum clot strength in vitro [31]. Venoms with relatively pure TLE activity exerted by serine proteases that do not engage factor XIII (FXIII) [42,43] result in a rapid onset of coagulation, but with a low velocity of clot growth and maximum clot strength in vitro [31,32]. Further, venoms with serine proteases that either act as a thrombin that polymerizes fibrinogen and activates FXIII or that engages other serine proteases to activate prothrombin and generate endogenous thrombin result in a significant decrease in time to onset of coagulation, increase in the velocity of clot growth, and increase in maximum clot strength in vitro [29,30,31,32]. As reviewed in [44,45], snake venom serine proteases have highly conserved features, such as histidine and aspartic acid residues within the catalytic site. Further, serine proteases also contain five disulfide bridges formed by ten cysteine residues [44,45]. The variability in glycosylation is primarily responsible for differences in serine protease activity, rendering them thrombin-generating or TLE in nature [44,45]. Critically, snake venoms, with their varied predominant enzymatic activities, subsequently inflict characteristic clinical and laboratory footprints in vivo in humans and in a rabbit model, as shown for *C. atrox* [46,47,48] and *C. adamanteous* [49,50,51]. In summary, kinetic and molecular investigations of medically important venoms are key to understanding their mechanism(s) of action and ultimately designing treatments to care for those affected by envenomation.

Thus, given the aforementioned factors, the present study investigating the medically important venom of *B. diporus* (BdipV) had the following goals: First, the kinetic behavior of *B. diporus* venom in human plasma was assessed with thromboelastography over a large range of concentrations and via spectrophotometry. Second, calcium-dependent and calcium-independent venom activities were determined. Third, we aimed to verify that serine proteases are the enzyme type primarily responsible for the venom activity observed. Lastly, we aimed to determine whether ruthenium chloride (RuCl_3_)-based antivenoms would have an effect on venom activity. These goals are outlined in Figure 1.

## 2. Results

### 2.1. Human Plasmatic Coagulation Concentration Response Determined by Thrombpelastography

In order to model the enhancement of coagulation in human plasma by BdipV, a model that allowed for an assessment of changes only due to venom activity and not endogenous thrombin generation was needed. Fortunately, when recalcified sodium-citrated anticoagulated plasma is placed into a thromboelastographic plastic cup/pin dyad, almost no measurable clot formation occurs for approximately 10 min, secondary to very weak contact protein-mediated (Factor XII) thrombin generation. Thus, the addition of a variety of concentrations of BdipV with data collected for only 10 min was expected to provide data attributable to action of the venom. This was found to be the case. Figure 2 displays the parameters measured using a plasma sample exposed to no BdipV (Control) and one exposed to 10 µg/mL of BdipV.

Human plasma was exposed to 0.01, 0.05, 0.1, 0.5, 1, 5, and 10 µg/mL concentrations of BdipV (N = 3 per concentration) and data were collected for 10 min. The effects of the venom on TMRTG, MRTG, and TTG values are presented subsequently in Figure 3, Figure 4 and Figure 5, respectively.

BdipV had a profound effect on coagulation kinetics in human plasma that plateaued between 1 and 10 µg/mL (Figure 3, Figure 4 and Figure 5). The adjusted R^2^ values of the fit of each model of thromboelastographic parameters were very good, with the percentage of the effects observed dependent on the concentration of BdipV, varying between 86% and 96%. Based on these results, subsequent thromboelastographic experiments utilized the largest concentration of BdipV tested (10 µg/mL) to ensure robust activity when assessing calcium dependence and vulnerability to endogenous antithrombin and inhibition by RuCl_3_-based antivenoms, as subsequently described.

### 2.2. Coagulant and Serine Protease Enzymatic Activities of BdipV on Citrated Human Plasma and a Chromogenic Substrate Cleaved by Serine Proteases

BdipV induced coagulation of citrated plasma in a dose-dependent manner. As the concentration increased, clotting time significantly decreased, reaching minimal values at the highest doses tested, as displayed in Figure 6. The minimum coagulant dose (MCD), defined as the amount of venom required to induce clotting within 60 s, was estimated to be approximately 2.97 µg. This profile suggests a marked serine protease procoagulant activity and a TLE activity that is calcium-independent, given that only fibrinogen can be polymerized in the absence of calcium.

Hydrolysis of the chromogenic substrate β-Ala-Gly-Arg-p-nitroanilide diacetate, a serine protease-specific chromogenic substrate commonly used to detect thrombin and thrombin-like enzyme activity, mediated by BdipV, displayed a dose-dependent response. Progressive p-nitroanilide release at 405 nm over an 18 min reaction period with increased concentrations of BdipV are displayed in Figure 7. Slopes calculated for each concentration revealed enzymatic kinetics proportional to the amount of venom used.

The association between coagulant activity (expressed as 1/clotting time) and enzymatic activity (based on the slope of p-nitroanilide release) was assessed. Pearson’s correlation analysis revealed an extremely strong positive correlation (r = 0.9925, *p* < 0.0001), with a 95% confidence interval ranging from 0.9479 to 0.9989. The coefficient of determination (R^2^ = 0.9850) indicates that 98.5% of the variability in coagulant activity can be explained by serine protease enzymatic activity measured in vitro. These results support the hypothesis that the observed procoagulant effect is directly related to a TLE, most likely a serine protease, present in BdipV.

### 2.3. Determination of Calcium Dependence of BdipV Activity, Identification of Thrombin-like Enzyme Activity, and Demonstration of Antithrombin-Mediated Inhibition with Thromboelastography

A variety of thromboelastographic methods were used to mirror the preceding biochemical methodologies to characterize BdipV activity. As displayed in Figure 8, within the 10 min of observation of recalcified plasma without addition of 10 µg/mL BdipV, very late-starting, slow-growing, weak clotting occurred. In contrast, BdipV in the presence of calcium induced a rapid onset of coagulation with a large velocity of growth and strong clot strength. Lastly, BdipV in the absence of calcium had significantly different coagulation kinetics from the other conditions, which is consistent with a venom with TLE activity.

In order to determine whether BdipV contained serine proteases responsible for the procoagulant activity observed, a series of experiments were conducted with the activation of the endogenous serine protease inhibitor, antithrombin, in the human plasma utilized. Calcium and 10 µg/mL BdipV were added to plasma exposed to 0, 5, and 10 U/mL unfractionated heparin, followed by thromboelastographic analyses for 10 min. As can be observed in Figure 9, progressive activation of endogenous antithrombin by heparin addition resulted in a significant heparin-concentration-dependent decrease in coagulation kinetics. This pattern of heparin-mediated inhibition is the sine qua non of plasma serine proteases.

To establish that the TLE activity noted in the absence of calcium observed in Figure 8 was most likely a serine protease, additional experiments were performed with plasma exposed to 0 or 5 U/mL of unfractionated heparin in plasma without calcium addition. Only 5 U/mL of heparin was utilized in these experiments given the weak coagulation elicited by the TLE in BdipV in Figure 8. As depicted in Figure 10, compared to plasma with no heparin addition, plasma with 5 U/mL of heparin addition had significantly less BdipV TLE-mediated coagulation. Thus, taken as a whole, these thromboelastographic analyses identified serine protease activity in BdipV in the presence and absence of calcium.

### 2.4. Effect of RuCl_3_-Based Antivenoms on BdipV Serine Protease-Mediated Procoagulant Activity

It has previously been demonstrated that a component of a Ru-based antivenom that is efficacious in vivo [52], RuCl_3_, inhibits the fibrinogenolytic activity of *C. atrox* venom in human plasma to a greater or lesser extent depending on the anionic composition of the solution and on whether the inorganic molecule dimethyl sulfoxide (DMSO) is present [53]. RuCl_3_ dissolved in phosphate-buffered solution inhibited the fibrinogenolytic activity of *C. atrox* venom to a significantly greater extent than RuCl_3_ in 0.9% NaCl (normal saline, NS) [53]. Solution containing RuCl_3_ in NS with 10% DMSO present (*v*/*v*) also inhibited *C. atrox* fibrinogenolytic activity to a significantly greater extent than RuCl_3_ in NS [51]. Thus, a series of experiments wherein BdipV was exposed in isolation to 100 µM RuCl_3_ in NS or phosphate buffer or NS with 10% DMSO for 5 min were performed. These three BdipV conditions were compared to BdipV not exposed to RuCl_3_. The results are displayed in Figure 11. BdipV serine protease activity was significantly inhibited by RuCl_3_ dissolved in NS compared to venom without RuCl_3_ exposure or exposure to RuCl_3_ dissolved in the two other solutions. These findings are in stark contrast to those obtained from testing *C. atrox* venom metalloproteinases [53]. In summary, the data in Figure 11 strongly supports the concept that only an ion formed by RuCl_3_ after loss of Cl is responsible for the inhibition of the serine protease activity of BdipV.

## 3. Discussion

Multiple methods of coagulation and spectrophotometric analysis allowed for the successful investigation of the kinetic patterns of and identification of the role of serine proteases associated with BdipV activity. The kinetic behavior of BdipV was determined using thromboelastography in plasma with the addition of calcium and via clotting time in plasma without calcium, with the finding that this venom was potent and contained thrombin-generating and TLE activities. The dependence on calcium was determined with both plasmatic coagulation kinetic methods in the process of kinetic modeling as well. Third, the finding that serine proteases are the enzyme type primarily responsible for the venom activity observed was confirmed spectrophotometrically by cleavage of β-Ala-Gly-Arg-p-nitroanilide diacetate, a serine protease-specific chromogenic substrate also hydrolyzed by thrombin-like enzymes and by BdipV. The results obtained demonstrate a strong link between the coagulant activity of BdipV on citrated human plasma and its ability to hydrolyze a serine protease-specific chromogenic substrate, widely used for both thrombin and thrombin-like enzyme activity. The extremely strong positive correlation (r = 0.9925) between the rate of p-nitroanilide release and the inverse of clotting time suggests that several TLE serine proteases in the venom are directly responsible for the observed procoagulant effect. Then, further validation of the presence of serine proteases that had both thrombin-generating and TLE activity was confirmed thromboelastographically by addition of heparin to activate endogenous plasmatic antithrombin, a serine protease inhibitor. Lastly, determination of inhibition of BdipV by various RuCl_3_ antivenoms was conducted, with a new pattern of compound-specific inhibition very different from the metalloproteases of *C. atrox* venom [53].

This type of procoagulant activity has been extensively reported in venoms from *Bothrops* species [29,30], where TLE serine proteases induce clot formation by converting fibrinogen into fibrin without activating the full coagulation cascade. The ability to predict biological effects (coagulation) from in vitro enzymatic assays provides a valuable functional tool for venom fraction characterization and for assessing their pathological potential. Furthermore, the determination of the MCD enables quantitative comparison of procoagulant potency, complementing the enzymatic profiling. The fact that both parameters are highly correlated supports the use of chromogenic assays as a simplified model for preliminary evaluation of thrombin-like activity in crude venoms or purified fractions. It is also critical to note that the venom utilized in the present investigation demonstrated activity remarkably different from that reported previously [30]. Those investigators found that *B. diporus* venom could not catalyze fibrinogen in the presence of calcium [30]. In sharp contrast, the present study demonstrated that *B. diporus* venom could polymerize fibrinogen in the presence or absence of calcium (Figure 8). The authors are very certain of the authenticity of the species from which the venom was obtained, given that the source is a primary repository of the viper within one of the countries afflicted by envenomation by *B. diporus*. Lastly, when combined with thromboelastographic methods, this multimodal approach to venom coagulation kinetic analysis can provide critical mechanistic insight.

Variation in inhibitory potency of RuCl_3_-based compounds between dominant venom enzyme activities is of great interest to those designing inorganic antivenoms. When comparing the proteomes of the venom of *C. atrox* and *B. diporus* (Table 1), it is apparent that the relative abundance of an enzyme type may have very little to do with the predominant nature of any particular venom. *C. atrox* venom has fibrinogenolytic, anticoagulant activity exerted by metalloproteinases [46,47,48], with nearly half the proteome composed of metalloproteinases (Table 1). In sharp contrast, the procoagulant venom of *B. diporus* contains slightly more than 7% serine proteases (Table 1), yet these enzymes markedly outcompete any other enzyme type in the proteome, as demonstrated by the present investigation. Therefore, given that venoms contain enzymes that can display high levels of activity, it is important to characterize venoms not only by relative abundance (which provides valuable evolutionary and clinical insights) but also by enzymatic activity. Incorporating functional assays such as clotting time, spectrophotometry, and thromboelastography offers a more complete characterization of venom activity. Critically, on the molecular level, serine proteases may well be vulnerable to Ru-based antivenoms, as Ru-containing compounds bind to histidine and sulfur-containing amino acids, as recently demonstrated and reviewed [53]. As reviewed, these amino acids are either in the catalytic site or are critical to serine protease activity [44,45]. Lastly, vulnerability to RuCl_3_-based venoms appears to be specific to venom enzyme type.

The present investigation has limitations. While human plasma was the primary medium used to characterize BdipV, it must be acknowledged that the effects of the venom on platelet activity (aggregation or inhibition of activation) were not investigated. While certainly of interest, the effects of the venom on coagulation kinetics in plasma needed to be determined first, especially with thromboelastography. If the kinetics of fibrin polymer matrix formation in plasma dynamically change in response to BdipV, then changes in platelet contributions to coagulation kinetics may be affected based on the characteristics of the plasma polymer matrix. The denser and stronger the fibrin polymer matrix, the stronger the effects of activated platelets may be as they bind to the matrix. Put another way, the ability of thromboelastography to assess the platelet contribution to whole-blood coagulation kinetics is dependent on the engagement to and contractile force of platelet attachments to the matrix. It would be better to expose isolated platelets to venom, wash them, and them place them into plasma not exposed to venom prior to analyses instead of interpreting data from whole blood without or with platelet inhibition exposed to venom. While important, these matters go beyond the scope of the present investigation, which had very focused goals. Future investigations of the effects of BdipV on platelet activity and additional characterizations of individual serine proteases isolated from the venom are justified.

In conclusion, the findings of the present study contribute to a better understanding of the functional and biochemical profile of BdipV and may have implications for the rational design of inhibitors, antivenom formulations, or preclinical models to study venom-induced coagulopathies.

## 4. Materials and Methods

### 4.1. Chemicals, Venom, and Plasma

#### 4.1.1. Chemicals

Calcium-free phosphate-buffered saline (PBS), RuCl_3_, dimethyl sulfoxide (DMSO), and thrombin-specific chromogenic substrate (β-Ala-Gly-Arg-p-nitroanilide diacetate, T3068) were obtained from Millipore Sigma (Millipore Sigma, Saint Louis, MO, USA). Unfractionated heparin (1000 U/mL) was obtained from Meitheal Pharmaceuticals, Inc., Chicago, IL, USA. Calcium chloride (200 mM) was obtained from Haemonetics Inc. (Braintree, MA, USA).

#### 4.1.2. Venom

Pooled *B. diporus* venom was obtained from adult specimens housed at the AGUARÁ Conservation Center, located in the province of Corrientes, in northeastern Argentina. Pooled venom was lyophilized and stored at −20 °C until use in laboratories at Consejo Nacional de Investigaciones Científicas y Técnicas. Additional venom samples were sent to the University of Arizona; they were dissolved in NS at a concentration of 36 mg/mL and stored at −80 °C until use.

#### 4.1.3. Plasma

Pooled normal human plasma that was sodium citrate-anticoagulated and maintained at −80 °C was obtained from George King Bio-Medical (Overland Park, KS, USA) prior to use at the University of Arizona. Human plasma was obtained from the blood of healthy volunteers of both sexes at the Applied Biochemistry Laboratory, Faculty of Exact and Natural Sciences and Surveying (FaCENA), National University of the Northeast (UNNE). Blood samples were collected by venipuncture using 3.2% sodium citrate (9:1, *v*/*v*) as an anticoagulant and then centrifuged at 1500× *g* for 10 min at room temperature. The resulting plasma samples were pooled prior to use. Pooled plasma that had never been frozen was utilized in the described experiment the same day it was collected. All experimental procedures were approved by the Ethics Committee of FaCENA-UNNE (Resolution No. 0011-23 CD), approval date: 1 January 2022.

### 4.2. Coagulation Activity

The coagulant activity of BdipV was determined according to the method described by Theakston and Reid [54] and Menaldo et al. [55], with slight modifications. Serial dilutions of BdipV (0.75–50 μg) were prepared in 50 µL of Tris buffer (pH 7.4) and incubated with 100 μL of citrated human plasma at 37 °C. Clotting time (in seconds) was measured using a Fibrintimer 2^®^ coagulometer (Wiener Lab, Santa Fe, Argentina). The minimum coagulant dose (MCD) was defined as the amount of venom capable of inducing plasma coagulation within 60 s. All experiments were performed in triplicate. The coagulant activity was evaluated by plotting protein concentration against clotting time.

### 4.3. Spectrophotometric Determination of Thrombin-like Activity

The thrombin-like activity of BdipV was evaluated as previously described by Kalainesan Rajesh Kumar et al. [56], with minor modifications. Briefly, the reactions were carried out in a final volume of 250 µL, containing 175 µL of Tris-HCl buffer (pH 7.4), the sample (BdipV, 50 µL), and 25 µL of 5 mM of the β-Ala-Gly-Arg-p-nitroanilide diacetate. The release of p-nitroanilide was continuously monitored at 405 nm in kinetic mode using a microplate reader for 18 min, and the standard curve was generated accordingly.

### 4.4. Thromboelastographic Analyses

The methodology for thromboelastographic analyses can be found in detail in previous works [31,32,52,53]. In brief, sample mixtures (always summating to 360 µL) were placed in a disposable cup in a computer-controlled thrombelastograph^®^ hemostasis system (Model 5000; Haemonetics Inc., Braintree, MA, USA) at 37 °C. The mixture used in the series of experiments was composed of 320 µL of plasma, 10–20 µL of PBS, 0–10 µL of BdipV in NS, and 0–20 µL of calcium chloride. BdipV was always added last to the plasma mixture to prevent any premature, unmonitored coagulation from occurring. The addition of vehicles or compounds to plasma or BdipV in isolation prior to experimentation are previously described in the individual series of experiments presented in the Results section. After BdipV addition and mixing by moving the cup up and down by the pin three times, data collection was commenced and thromboelastographic parameter values as displayed in Figure 1 were collected for 10 min.

### 4.5. Statistical Analyses

#### 4.5.1. Analyses Conducted at Consejo Nacional de Investigaciones Científicas y Técnicas

Quantitative data were shown as the mean ± SE of the standard error of the mean (SEM) of at least three independent experiments. Additionally, the correlation between coagulant activity and thrombin-like activity was analyzed using GraphPad Prism v8.0.2 (GraphPad Software, San Diego, CA, USA). The analysis was based on slope values obtained from the linear regression of kinetic curves and on coagulation data expressed as the inverse of time (1/time, in seconds). The association between variables was assessed using Pearson’s correlation coefficient, and model fit was reported using the coefficient of determination (R^2^). A *p* value < 0.05 was considered statistically significant.

#### 4.5.2. Analyses Conducted at the University of Arizona

Data are presented mean + standard deviation (SD) or as raw data as part of modeling of kinetic behavior determined with thromboelastography. Data modeling coagulation kinetics utilized a commercially available graphics program (Origen 2025, OrigenLab Corporation, Northampton, MA, USA), with N = 3 per concentration of venom tested. In all other experiments, each condition was represented by N = 6 replicates, as this provided a statistical power ≥0.8, with *p* < 0.05; this methodology was used to assess differences in thromboelastographic parameters [31,32,52,53]. A commercially available statistical program was used for unpaired Student’s t-test or one-way ANOVA, followed by Holm–Sidak post hoc analyses as appropriate (SigmaStat 3.1; Systat Software, Inc., San Jose, CA, USA). *p* < 0.05 was considered significant. Graphics were generated with the commercially available programs Origen 2025 and CorelDRAW 2024, Alludo, Ottawa, ON, Canada.

## Figures and Tables

**Figure 1 ijms-26-09496-f001:**
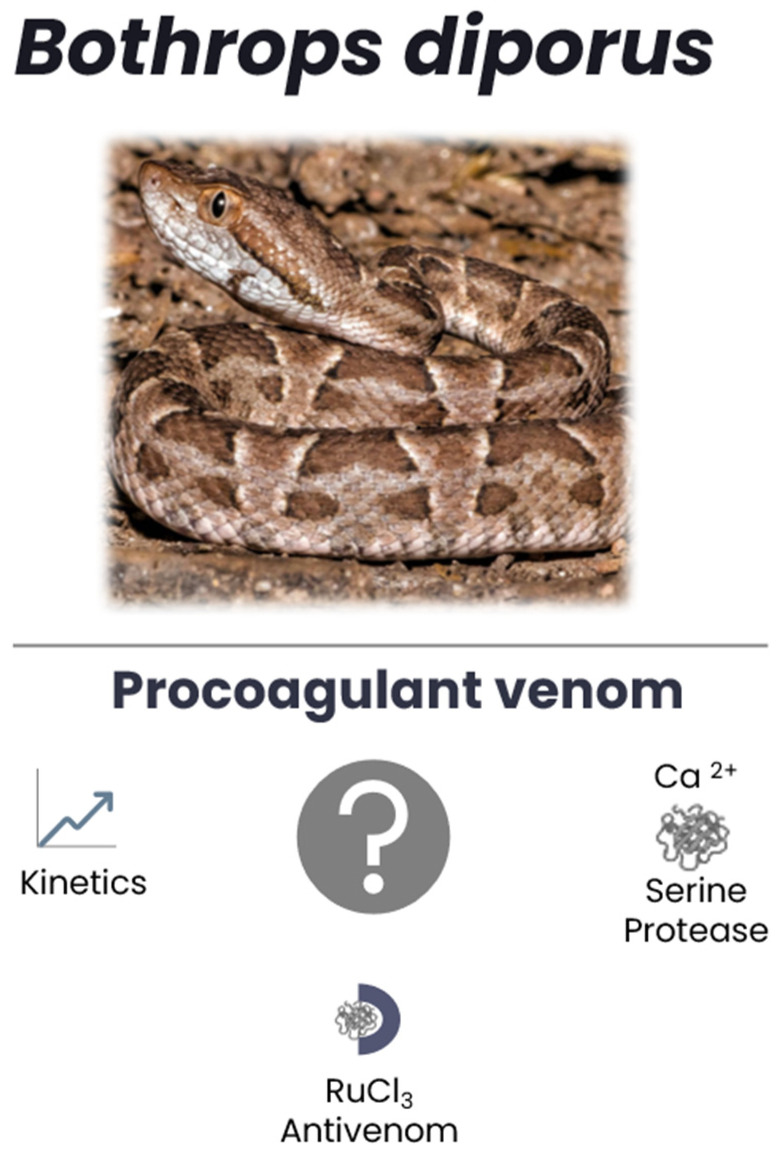
Study goals of investigation of BdipV, a presumed prothrombotic venom. Coagulation kinetics, calcium-dependent activity, identity of serine protease activity, and vulnerability to RuCl_3_-based antivenoms were to be determined. Photo courtesy of Hernan Rojo.

**Figure 2 ijms-26-09496-f002:**
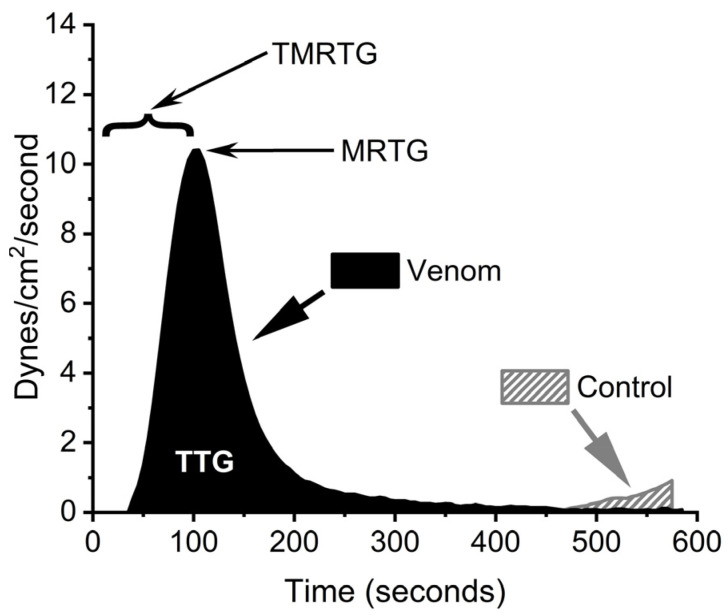
Paradigm to assess kinetics of BdipV in human plasma. Within a 10 min period, there is almost no endogenous clot formation (Control, hatched trace). With venom (BdipV, 10 µg/mL, black trace), the clot formed before endogenous clotting has commenced. TMRTG = time to maximum rate of thrombus generation (minutes or seconds), a measure of commencement of coagulation; MRTG = maximum rate of thrombus generation (dynes/cm^2^/second), a measure of the velocity of clot growth; and TTG = total thrombus generation (dynes/cm^2^), a measure of clot strength.

**Figure 3 ijms-26-09496-f003:**
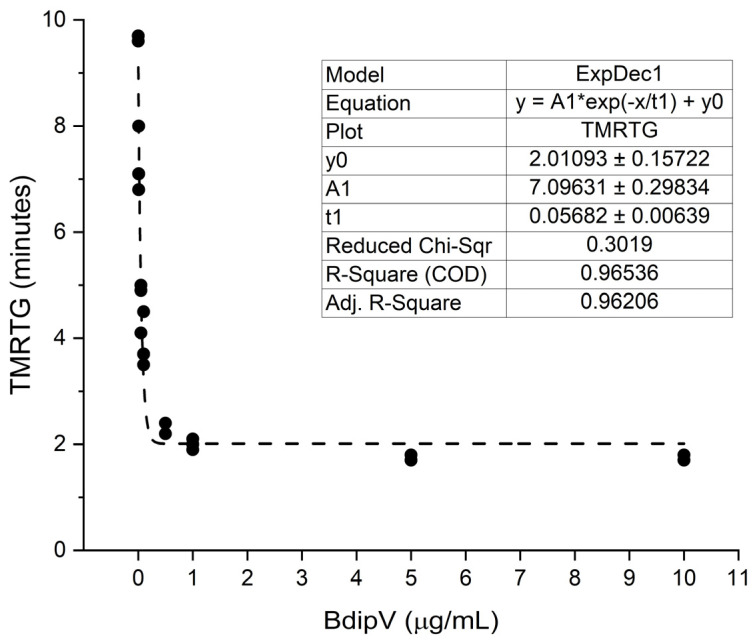
Effects of BdipV on TMRTG in human plasma. TMRTG demonstrated an exponential decay in values as venom concentration increased. N = 3 per concentration.

**Figure 4 ijms-26-09496-f004:**
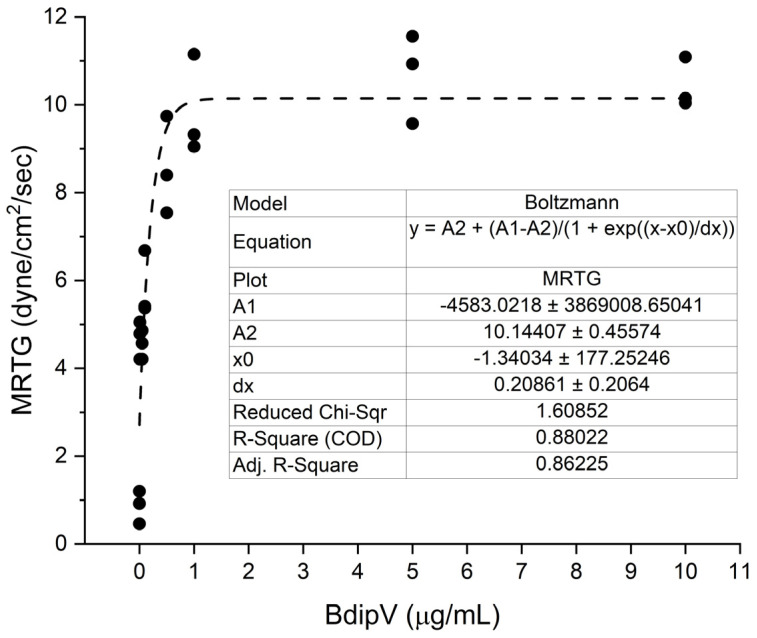
Effects of BdipV on MRTG in human plasma. MRTG demonstrated an increase in values best modeled as a Boltzmann relationship to increases in venom concentration. N = 3 per concentration.

**Figure 5 ijms-26-09496-f005:**
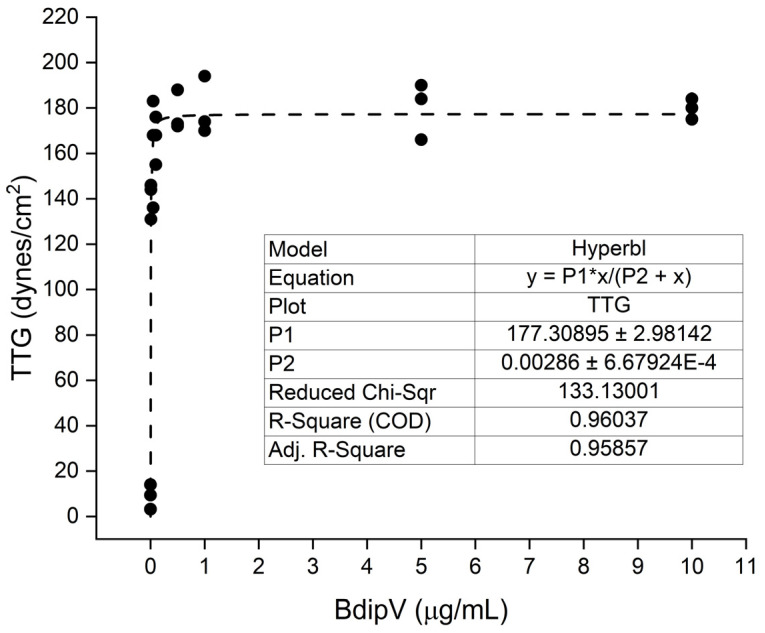
Effects of BdipV on TTG in human plasma. TMRTG demonstrated a hyperbolic increase in values as venom concentration increased. N = 3 per concentration.

**Figure 6 ijms-26-09496-f006:**
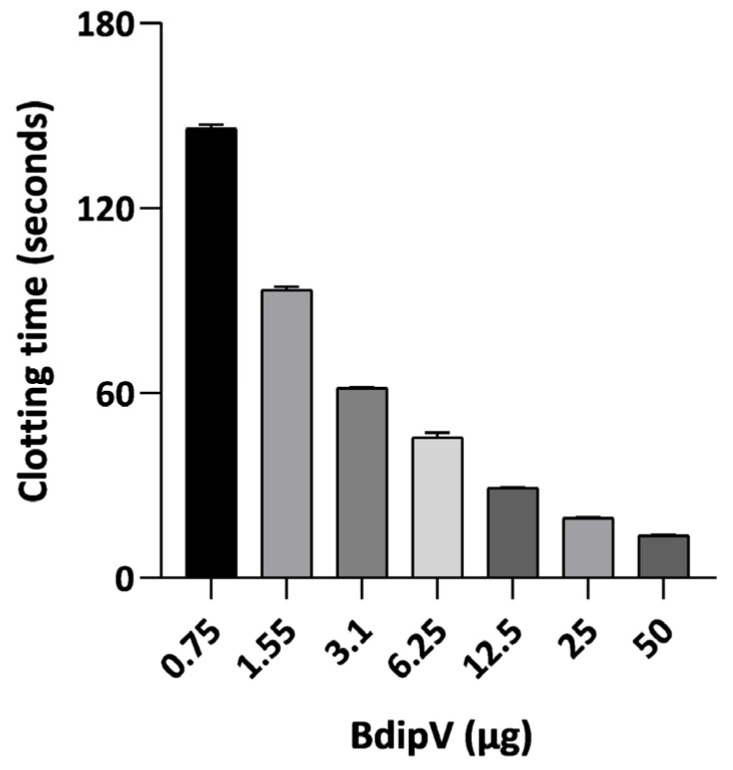
Concentration response data for clotting time elicited by BdipV in citrated human plasma. Data are expressed as mean ± standard error of the mean (SE, N = 3 per concentration).

**Figure 7 ijms-26-09496-f007:**
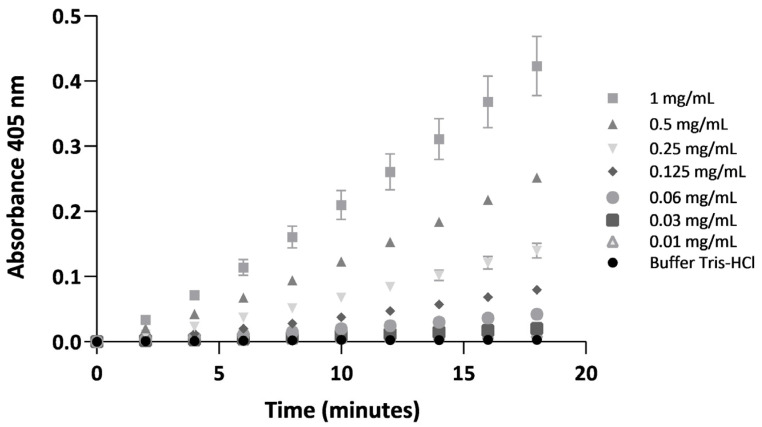
Progressive catalysis of the serine protease-specific chromogenic substrate β-Ala-Gly-Arg-p-nitroanilide diacetate by increasing concentrations of BdipV in Tris-HCl buffer. The release of p-nitroanilide at 405 nm was monitored over 18 min, and the slope of absorbance increase was calculated for each venom concentration. Data are expressed as mean ± SE, N = 3 per concentration.

**Figure 8 ijms-26-09496-f008:**
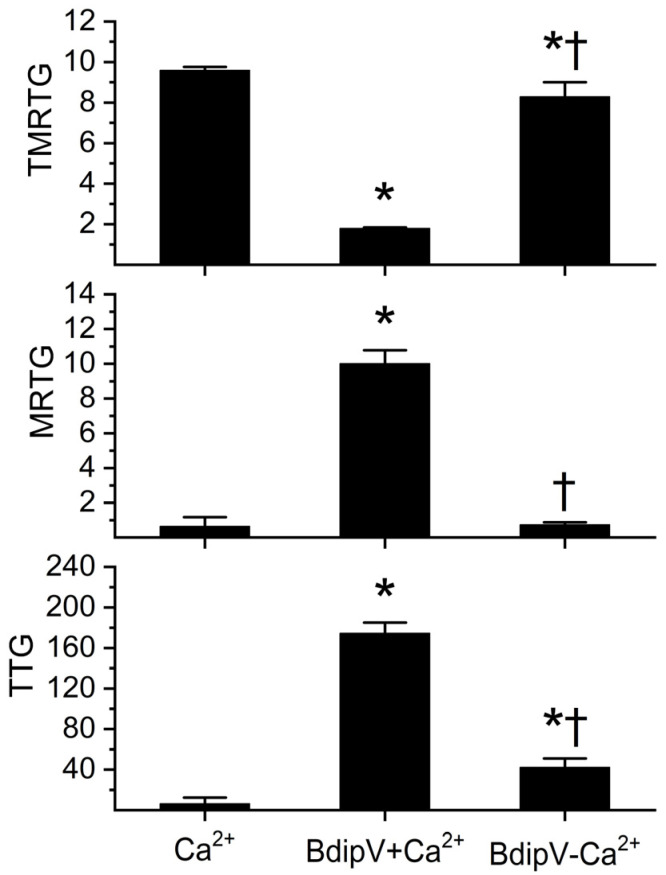
Impact of BdipV on coagulation kinetics in the presence and absence of calcium. Ca^2+^ = plasma with calcium addition without BdipV; BdipV + Ca^2+^ = plasma with calcium addition and 10 µg/mL BdipV; and BdipV-Ca^2+^ = plasma without calcium with 10 µg/mL BdipV. Data were analyzed with one-way ANOVA with Holm–Sidak post hoc test. * *p* < 0.05 vs. Ca^2+^; † *p* < 0.05 vs. BdipV + Ca^2+^. N = 6 per condition.

**Figure 9 ijms-26-09496-f009:**
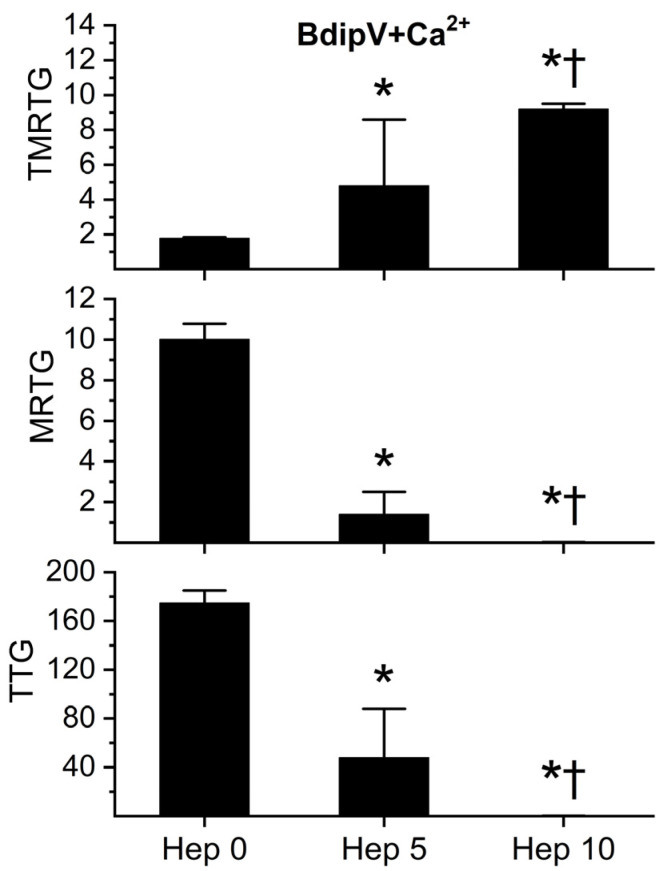
Impact of addition of heparin on BdipV on coagulation kinetics in the presence of calcium. BdipV + Ca^2+^ = plasma with calcium addition and 10 µg/mL BdipV for all three conditions; Hep 0 = no heparin addition; Hep 5 = 5 U/mL addition; and Hep 10 = 10 U/mL heparin addition. Data were analyzed with one-way ANOVA with Holm–Sidak post hoc test. * *p* < 0.05 vs. Hep 0; † *p* < 0.05 vs. Hep 5. N = 6 per condition.

**Figure 10 ijms-26-09496-f010:**
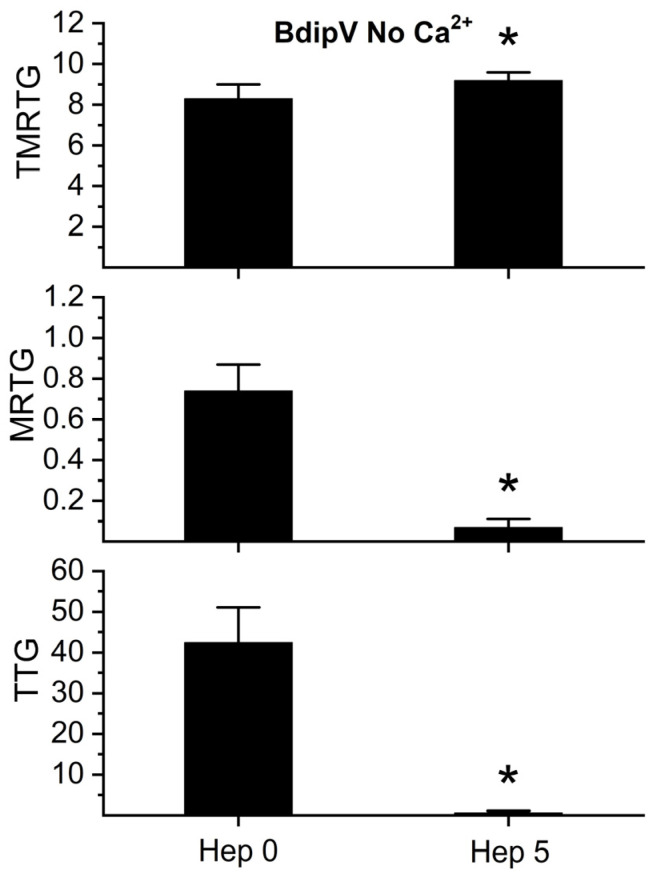
Impact of addition of heparin on BdipV on coagulation kinetics in the absence of calcium. BdipV No Ca^2+^ = plasma without calcium addition and 10 µg/mL BdipV for all three conditions; Hep 0 = no heparin addition; Hep 5 = 5 U/mL addition. Data were analyzed with unpaired, two-tailed Student’s *t* test. * *p* < 0.05 vs. Hep 0. N = 6 per condition.

**Figure 11 ijms-26-09496-f011:**
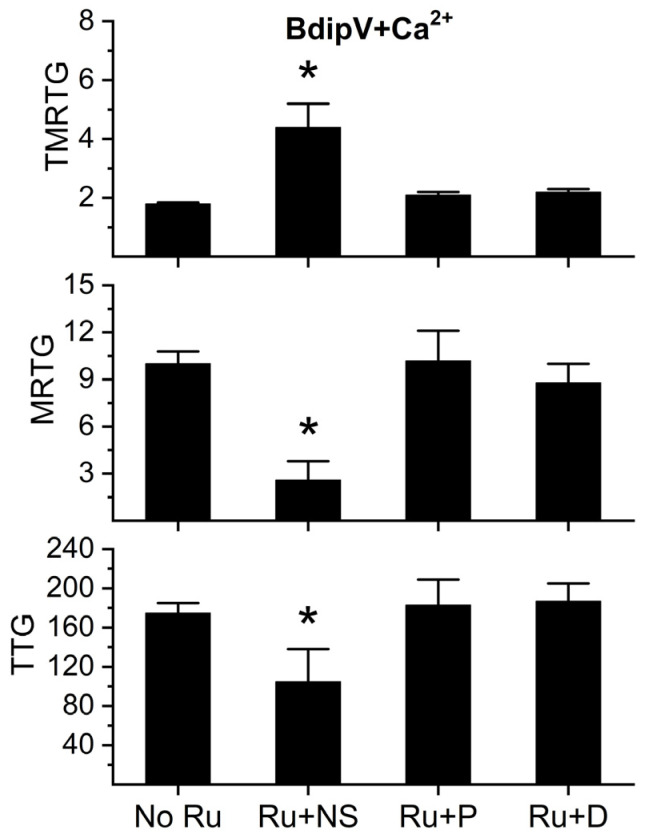
Differential inhibition of BdipV serine protease activity by RuCl_3_. BdipV + Ca^2+^ = plasma with calcium addition and 10 µg/mL BdipV for all four conditions; No Ru = BdipV not exposed to RuCl_3_; Ru + NS = BdipV exposed to 100 µM RuCl_3_ dissolved in NS; Ru+ P = BdipV exposed to 100 µM RuCl_3_ dissolved in phosphate-buffered saline; and Ru + D = BdipV exposed to 100 µM RuCl_3_ dissolved in NS with 10% DMSO addition (*v*/*v*). Data were analyzed with one-way ANOVA with Holm–Sidak post hoc test. * *p* < 0.05 vs. all other conditions. N = 6 per condition.

**Table 1 ijms-26-09496-t001:** Percent relative abundance of three venom enzymes in hemotoxic snake venoms.

Venom Enzyme	*C. atrox* [37]	*C. adamanteous* [38]	*B. diporus* [39]
Metalloproteinase	49.7%	24.4%	34.2%
Serine Protease	19.8%	20%	7.2%
Phospholipase A_2_	7.3%	7.8%	24.1%

## Data Availability

The original contributions presented in this study are included in this article. Further inquiries can be directed to the corresponding author.

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
