# Peer review of "Procoagulant Effects of Bothrops diporus Venom: Kinetic Modeling and Role of Serine Protease Activity"

_ijms, 2025, doi:10.3390/ijms26199496_

Round 1

Reviewer 1 Report

Comments and Suggestions for Authors

This study provides a preliminary characterization of a snake venom. Using several standard assays, the procoagulant activity of the venom has been demonstrated, which is likely mediated by the serine protease and thrombin-like enzyme activities. The topic is of potential interest to the animal toxin community and could, in the long term, contribute to the design of antivenoms or therapeutic inhibitors. While the manuscript is clearly written and the experimental approach is fundamentally sound, the study currently lacks the mechanistic depth and rigorous validations/characterizations, which are required for publication in IJMS. The data presented are largely descriptive and do not significantly advance our understanding beyond the initial observation. There are several major issues:

  • The entire study characterizes the activity of a complex, whole venom mixture rather than a defined/purified toxin. While enzymatic assays provide a functional overview, they do not elucidate the concentrations, molecular properties, specific activities, or potential synergistic/antagonistic interactions of the individual active ingredients within the venom. This fundamental limitation significantly reduces the mechanistic insight and overall impact of the work. Considering several key citations, the descriptive nature of the study, which builds upon well-established methodologies, may be better suited for a more specialized MDPI journal such as Toxins.
  • The current presentation of Fig 2 is very unprofessional.
  • Suggest to include error bars (e.g., SD or SEM) and using the logarithmic values for the x-axis in Fig 3/4/5, to better represent the dose-response relationship.
  • Suggest to show individual data points (e.g., scatter plots overlaying the bars) in Fig 6/8/9/10/11.
  • In Fig 7, the data for the 0.5 mg/mL group, which is in the middle of 1 and 0.25 mg/mL groups, are anomalous. Very small error bars are showing. This requires experimental verification and explanation.
  • The design and reporting of the calcium-dependent experiments (e.g., Fig. 8) are unclear and critically limit interpretation. What’s the concentration of Ca2+ has been used in the assay? “+Ca2+” means adding Ca2+ directly into plasma? How the influence of the basal calcium level in the plasma was controlled for or accounted for? Furthermore, for all statistical comparisons, significance must be explicitly indicated on the figures with connecting lines.

Minor issues:

  • Lines 23-24: “activity” has been mentioned twice. Suggest to revised as “Both calcium-dependent and calcium-independent activities were observed…”
  • Line 268: “determine” should be “determined”.
  • Lines 274-275: “hydrolyze a” has been typed twice.

Author Response

“This study provides a preliminary characterization of a snake venom. Using several standard assays, the procoagulant activity of the venom has been demonstrated, which is likely mediated by the serine protease and thrombin-like enzyme activities. The topic is of potential interest to the animal toxin community and could, in the long term, contribute to the design of antivenoms or therapeutic inhibitors. While the manuscript is clearly written and the experimental approach is fundamentally sound, the study currently lacks the mechanistic depth and rigorous validations/characterizations, which are required for publication in IJMS. The data presented are largely descriptive and do not significantly advance our understanding beyond the initial observation. There are several major issues:” We disagree that our work, which includes “preliminary” in the title, did not use mechanistic approaches to define the likely enzymes responsible for the procoagulant activity recorded. The use of specific spectrophotometric substrates, antithrombin mediated inhibition, calcium modulation, and ruthenium exposure provided mechanistic data well worthy of a journal such as IJMS.

“The entire study characterizes the activity of a complex, whole venom mixture rather than a defined/purified toxin.” That is correct. But it is only by testing the whole venom that it can be discerned which specific enzymes are responsible for the phenomena observed clinically, manifesting as profound coagulopathy.

“While enzymatic assays provide a functional overview, they do not elucidate the concentrations, molecular properties, specific activities, or potential synergistic/antagonistic interactions of the individual active ingredients within the venom.”  As explicitly stated, we wished to learn how this whole venom behaved as a procoagulant in human plasma and then determine if serine protease activity was responsible for the data observed. Additional determinations of the degree of action based on calcium presence or ruthenium inhibition were also sought.

“This fundamental limitation significantly reduces the mechanistic insight and overall impact of the work. Considering several key citations, the descriptive nature of the study, which builds upon well-established methodologies, may be better suited for a more specialized MDPI journal such as Toxins.” Of the many manuscripts published by the corresponding author in both IJMS and Toxins, all involved utilization of the thrombelastograph with a variety of modifications to answer questions about venom and its interaction with plasma and blood in vitro and in vivo. The mechanistic insights gained were to be elucidated with the experimental designs presented in these works over the past few years.  Our spectrophotometric and modified thrombelastographic work is just descriptive – it is diagnostic for serine protease activity.

“The current presentation of Fig 2 is very unprofessional.” We are surprised at this comment, as similar explanations of the first derivative variables of thrombelastography have appeared in not just multiple hematological journals over 20 years but also IJMS and Toxins. The figure clearly conveys not only the critical aspects of timing of the assay, wherein the rapid onset of coagulation caused by the venom occurs, but also identifies the variables of interest for the readership. 

“Suggest to include error bars (e.g., SD or SEM) and using the logarithmic values for the x-axis in Fig 3/4/5, to better represent the dose-response relationship.” These figures provide the raw modeling data – which are clearly not appropriate for logarithmic analyses – along with the precise model, goodness of fit, and approximation of relevant constants. Provision of the raw data, not SD or SEM, is the most complete manner in which to present these analyses. exponential decay, Boltzmann, and hyperbolic models fit these data extraordinarily well. 

“Suggest to show individual data points (e.g., scatter plots overlaying the bars) in Fig 6/8/9/10/11.” We are again confused by a need to change our graphics for the experimental design and analyses depicted. Traditional measures of central tendency and variance are routinely displayed with bar graphics. Our data is unambiguous and strongly support our conclusions. 

“In Fig 7, the data for the 0.5 mg/mL group, which is in the middle of 1 and 0.25 mg/mL groups, are anomalous. Very small error bars are showing. This requires experimental verification and explanation.”  The reviewer may be expecting that increased variance is always expected with increasing concentrations of enzyme or substrate. That is simply not the case – these are experimental data. The replicates were represented by n=3. The pattern of catalysis is consistent and dependent on substrate concentration. They hypothesis tested involves the concentration-dependent pattern of catalysis, not individual variability of each concentration of substrate. 

“The design and reporting of the calcium-dependent experiments (e.g., Fig. 8) are unclear and critically limit interpretation. What’s the concentration of Ca2+ has been used in the assay? “+Ca2+” means adding Ca2+ directly into plasma? How the influence of the basal calcium level in the plasma was controlled for or accounted for?” Sodium citrate anticoagulated plasma has a near zero calcium concentration – allowing for the conduct of calcium free conditions to experiment that can be contrasted to the addition of exogenous calcium to restore normal concentrations to discern the activity of calcium-dependent activity. The concentration of calcium using the constituents described is normal and has been used in hundreds of manuscripts utilizing thrombelastography and citrated blood products. This methodology of using sodium citrate to anticoagulated samples and then add back calcium to assess coagulation function is nearly a century old. As indicated, conditions with “+Ca2+” means calcium addition to the sample.

“Furthermore, for all statistical comparisons, significance must be explicitly indicated on the figures with connecting lines.”  Our representation of the data are very mainstream and we do not wish to detract from the clarity of our figures.

Minor issues:

“Lines 23-24: “activity” has been mentioned twice. Suggest to revised as “Both calcium-dependent and calcium-independent activities were observed…”  We have modified the sentence.

“Line 268: “determine” should be “determined”. We have made the change.

“Lines 274-275: “hydrolyze a” has been typed twice.” We have deleted the repeated phrase.

Reviewer 2 Report

Comments and Suggestions for Authors

The present study addresses a clinically relevant and biologically significant topic: the characterization of procoagulant activity in Bothrops diporus venom, a major cause of envenomation and coagulopathy in Argentina. Through a multimodal approach,including clotting time assays, spectrophotometry, and thrombelastography, the authors demonstrate that B. diporus venom exhibits potent thrombin-generating and thrombin-like enzymatic activity, largely attributable to serine proteases. The findings are further supported by calcium-dependence profiling and inhibition assays using heparin and RuCl₃, which reinforce the mechanistic role of serine proteases in venom-induced coagulation. However, the manuscript requires substantial attention to language, structure, and clarity. Numerous typographical errors, inconsistent phrasing, and long sentence constructions hinder comprehension. In summary, the study presents important and novel findings, but its readability and presentation must be improved to better communicate its contribution to the field.

  1. Please rephrase this sentence: The authors sought to define the molecular toxin responsible for coagulopathy with several methods utilized by the two laboratories. This sounds like a technical report, instead of a manuscript.
  2. Line 23. This is redundant and repetitive: calcium dependent activity and calcium independent activity identified
  3. Lines 23-24. Please the repetition of words. Activity has been used 4 times in two lines.
  4. Figure 1 can be added as supplementary material or graphical abstract, but it does not add value to the current version of the manuscript.
  5. Line 58. Bothrops must be italicised.
  6. Line 55. The authors must revise this sentence. Thromboelastography has been employed for more than two groups, including Guiterrez’s team.
  7. This sentence does not contribute to the introduction: The source of the venom was stated to be Venom Evolution Lab, UQ, Australia; the venom came from adult  snakes, but the precise locality of origin was unknown, and it was unknown if the venom came from an individual viper or was part of a pooled sample.
  8. The authors have mentioned that The thrombelast graphic footprint of B. diporus [30] is consistent with serine protease containing venoms that either act as thrombin or generate thrombin without the ability to directly polymerize fibrinogen as a thrombin-like enzyme (TLE). But the introduction does not include a description of serinoproteases. The introduction must be enriched with more information of this toxin family. I suggest some reviews to enrich this aspect of the introduction, DOI: 3390/biom15020154 and DOI: 10.1016/j.toxicon.2012.09.003
  9. Lines 88-90. A reference must be used to support this statement.
  10. Lines 88-90. This can be an extrapolation without evidence, as the use of synthetic substrates for measuring kinetic parameters does not reflect the natural environment. In addition, some pharmacological properties of venom components can be dissociated from catalytic activity, as have been well-documented for PLA2s.
  11. Personal expectations should be not included in the manuscript.
  12. Figures 3, 4 and 5 can be combined in just one panel.
  13. The manuscript contains many figures, which can be combined for better understanding.
  14. Line 265. This should not be included in the manuscript.
  15. Sentence structure is often dense and repetitive. Please check the entire document.
  16. Some parts of the document needs extensive English editing, as the meaning is not clear. For example: hydrolyze a hydrolyze,” “aggregation of inhibition of activation”, and so on.
  17. The sentence “serine proteases are the enzyme type primarily responsible” could be strengthened by specifying which serine proteases were identified or suspected, especially given the proteomic data.
  18. The discussion of RuCl₃ inhibition is underdeveloped. It would benefit from more mechanistic speculation or reference to known coordination chemistry principles that might explain the selective inhibition.
  19. The strong correlation (r = 0.9925) between chromogenic substrate hydrolysis and clotting time was demonstrated, but correlation does not imply causation. The authors should acknowledge this and consider whether other factors (e.g., non-serine protease components) might contribute to clotting kinetics.
  20. The omission of platelet function analysis is significant, especially given the emphasis on thrombelastography. The authors mentioned this, but the rationale for excluding it (“needed to be performed first”) feels weak. Could platelet assays have been run in parallel?
  21. Were any efforts made to distinguish between thrombin-like enzymes with fibrinogen-clotting activity and those with broader proteolytic profiles? Could substrate specificity assays help
  22. How do the findings translate to envenomation scenarios? Is the procoagulant activity of BdipV consistent with clinical manifestations in bite victims, or are there compensatory anticoagulant components in the whole venom?
  23. Could the chromogenic substrate used be cleaved by non-TLE proteases in the venom? How was specificity ensured? 

Author Response

“The present study addresses a clinically relevant and biologically significant topic: the characterization of procoagulant activity in Bothrops diporus venom, a major cause of envenomation and coagulopathy in Argentina. Through a multimodal approach, including clotting time assays, spectrophotometry, and thrombelastography, the authors demonstrate that B. diporus venom exhibits potent thrombin-generating and thrombin-like enzymatic activity, largely attributable to serine proteases. The findings are further supported by calcium-dependence profiling and inhibition assays using heparin and RuCl₃, which reinforce the mechanistic role of serine proteases in venom-induced coagulation. However, the manuscript requires substantial attention to language, structure, and clarity. Numerous typographical errors, inconsistent phrasing, and long sentence constructions hinder comprehension. In summary, the study presents important and novel findings, but its readability and presentation must be improved to better communicate its contribution to the field.”

“Please rephrase this sentence: The authors sought to define the molecular toxin responsible for coagulopathy with several methods utilized by the two laboratories. This sounds like a technical report, instead of a manuscript.” We appreciate this comment and have restructured the sentence.

“Line 23. This is redundant and repetitive: calcium dependent activity and calcium independent activity identified” This has been modified.

“Lines 23-24. Please the repetition of words. Activity has been used 4 times in two lines.” We have revised the text to reduce the repetition of the word ‘activity’ in lines 23–24.

“Figure 1 can be added as supplementary material or graphical abstract, but it does not add value to the current version of the manuscript.”  The graphic prepares the interested reader for what is about to be presented. We prefer to keep the figure.

“Line 58. Bothrops must be italicised.” The text has been modified.

“Line 55. The authors must revise this sentence. Thromboelastography has been employed for more than two groups, including Guiterrez’s team.” Thrombelastography has been used by more than two groups, but two have extensively focused on Bothrops species to the extent that we and the Fry’s group has. I cannot find a reference with Guiterrez and thrombelastography or Rotem and Bothrops venom. We have modified the text to not exclude other groups.

“This sentence does not contribute to the introduction: The source of the venom was stated to be Venom Evolution Lab, UQ, Australia; the venom came from adult  snakes, but the precise locality of origin was unknown, and it was unknown if the venom came from an individual viper or was part of a pooled sample.” Given the rest of the text surrounding this statement, this sentence definitely contributes to the Introduction. Even more critically, the well demonstrated catalysis of fibrinogen demonstrated by our data is in stark contrast to Fry’s data, making the statement particularly relevant. Further, Fry’s thrombelastogram of B. diporus venom mediate coagulation does not look like what we found, and this may very well be secondary to where the individual or pooled venom was collected by the Fry group. We are very specific about where our venom was obtained.

“The authors have mentioned that The thrombelast graphic footprint of B. diporus [30] is consistent with serine protease containing venoms that either act as thrombin or generate thrombin without the ability to directly polymerize fibrinogen as a thrombin-like enzyme (TLE). But the introduction does not include a description of serinoproteases. The introduction must be enriched with more information of this toxin family. I suggest some reviews to enrich this aspect of the introduction, DOI: 3390/biom15020154 and DOI: 10.1016/j.toxicon.2012.09.003” We cannot locate the first reference but now cite the second review. We have provided more information concerning serine proteases as requested in the Introduction, and the second reference was particularly useful when explaining Ru mediated inhibition of activity on the molecular level.

“Lines 88-90. A reference must be used to support this statement.” The statement is a basic precept of enzyme kinetics. The need to cite a basic textbook concerning the subject should not be needed. The readership should have a basic understanding of enzyme kinetic principles to be able to understand the content of our manuscript. We have modified the sentence as follows: “Specifically, if the Michaelis constant (Km) of a snake venom enzyme (or any other enzyme) is small and the maximum rate of reaction (Vmax) is large, then it will outcompete other venom enzymes present that catalyze the same substrate, resulting in a and display a characteristic footprint in terms of assessment parameters”.

“Lines 88-90. This can be an extrapolation without evidence, as the use of synthetic substrates for measuring kinetic parameters does not reflect the natural environment. In addition, some pharmacological properties of venom components can be dissociated from catalytic activity, as have been well-documented for PLA2s.” Our work combines the use of human plasma, a natural environment, to complement spectrophotometric studies in buffer. The data generated in plasma and the conclusions drawn from the data are evidence-based.

“Personal expectations should be not included in the manuscript.” Outcomes that are expected based on basic enzyme kinetic principles are not a matter of personal desire.

“Figures 3, 4 and 5 can be combined in just one panel.

The manuscript contains many figures, which can be combined for better understanding.” The ability to examine the raw data and read the modeling program data in the embedded table within figures 3-5 would be grossly compromised by combining them. As an editor and reviewer for the journal, the corresponding author prefers the approach of separating multipaneled figures as they are unreadable and become so complicated that the readership will not appreciate the work.

“Line 265. This should not be included in the manuscript.” We are presuming the simple declarative sentence “The present study achieved its stated goals”. is the problem. We have replaced the sentence.

“Sentence structure is often dense and repetitive. Please check the entire document.

Some parts of the document needs extensive English editing, as the meaning is not clear. For example: hydrolyze a hydrolyze,” “aggregation of inhibition of activation”, and so on.” We have modified the phrases indicated and checked the entire document.

“The sentence “serine proteases are the enzyme type primarily responsible” could be strengthened by specifying which serine proteases were identified or suspected, especially given the proteomic data.” This comment is confusing, as we clearly did not isolate and purify each individual serine protease possible, characterize them, and present primary proteomic data. Our data clearly identify serine proteases as responsible for the phenomena observed in the various experimental series. 

“The discussion of RuCl₃ inhibition is underdeveloped. It would benefit from more mechanistic speculation or reference to known coordination chemistry principles that might explain the selective inhibition.” The purpose of this series of experiments was to identify what, if any, RuCl3 species affect venom activity. While the identification of various amino acid targets of Ru have been reviewed [reference 53], a more detailed discussion beyond what is already presented would be very speculative. The identification of which venom enzyme amino acid residue is bound by RuCl3, the molecular conformational changes of the enzyme that occur documented via crystallography, and in vitro enzyme-substrate interactions determined spectrophotometrically are subjects for future investigation.

“The strong correlation (r = 0.9925) between chromogenic substrate hydrolysis and clotting time was demonstrated, but correlation does not imply causation. The authors should acknowledge this and consider whether other factors (e.g., non-serine protease components) might contribute to clotting kinetics.” As stated in Results: “The coefficient of determination (R² = 0.9850) indicates that 98.5% of the variability in coagulant activity can be explained by the serine protease enzymatic activity measured in vitro”. This means that 1.5% of the observations are explainable by enzymes/proteins that are not serine proteases. Correlation does not prove (or imply) causation when considered in abstraction. However, if the one thing that can cleave the serine protease-substrate in the spectrophotometric assay correlates with what happens in human plasma except for 1.5% of the observations, then causation is as definitively proven as is humanly possible.

“The omission of platelet function analysis is significant, especially given the emphasis on thrombelastography. The authors mentioned this, but the rationale for excluding it (“needed to be performed first”) feels weak. Could platelet assays have been run in parallel?” As the reviewer knows, platelets have several receptor and post receptor activations systems that are activated by multiple compounds that include proteins and enzymes. It is certain that whatever is activating/inhibiting platelets in whole venom would not be limited to serine proteases. In sharp contrast, the serine protease, FXIII, and fibrinogen activation by serine proteases is easily identified with the methods presented in our manuscript.

“Were any efforts made to distinguish between thrombin-like enzymes with fibrinogen-clotting activity and those with broader proteolytic profiles? Could substrate specificity assays help.” The experiments that involved omission of calcium definitively demonstrate TLE activity, as the generation of thrombin is calcium dependent. The robust response of the enzyme in plasma with calcium present are characteristic of FXIII activation, which is dependent on calcium generation and thrombin generation. No further experiments would provide more information concerning the presence of TLE activity.

“How do the findings translate to envenomation scenarios? Is the procoagulant activity of BdipV consistent with clinical manifestations in bite victims, or are there compensatory anticoagulant components in the whole venom?” The in vitro findings of the present study are consistent with the regional thrombosis or rapid systemic loss of fibrinogen accompanied by catastrophic hemorrhage. We have cited several studies in the manuscript that substantiate this conclusion.

“Could the chromogenic substrate used be cleaved by non-TLE proteases in the venom? How was specificity ensured?” The substrate, b-Ala-Gly-Arg p-nitroanilide diacetate, is a well-characterized, specific substrate for serine proteases. The substrate specified in the kit we used as one that can be used in thrombin generation assays, and is therefore considered specific for thrombin-like enzymes (TLEs). In our experiments, the enzyme in the venom, being a serine protease with TLE activity, acted directly on this substrate. According to the kit instructions, we simply incubated the substrate with the venom containing the TLE, supporting the specificity of the reaction.

Round 2

Reviewer 1 Report

Comments and Suggestions for Authors

I'm not satisified with the response.

Author Response

"I'm not satisified with the response." We responded to the comments of the reviewer in the first round to the best of our ability. In the second round, the reviewer has not acknowedged our efforts and provides no specific matter to address. Thus, we respectfully appreciate the input of the reviewer in the revision of our work that will be submitted.

Reviewer 2 Report

Comments and Suggestions for Authors

The authors have addressed the main points, although I believe the manuscript could be improved by presenting panels of figures with consistent context. Despite these considerations, I believe the manuscript makes a valuable contribution to the literature and advances our understanding of the actions of Bothrops diporus venom. The authors have also replied all comments. 

Author Response

"The authors have addressed the main points, although I believe the manuscript could be improved by presenting panels of figures with consistent context. Despite these considerations, I believe the manuscript makes a valuable contribution to the literature and advances our understanding of the actions of Bothrops diporus venom. The authors have also replied all comments." We appreciate the reviewer's comment concerning our figures, but give the diversity of method and interpretation of our data, the figures are as consistent as we can make them. We also appreciate the reviewer's kind comment concerning the value of our work and the acknowledgement that we have addressed all comments.